# Piperine and Its Metabolite’s Pharmacology in Neurodegenerative and Neurological Diseases

**DOI:** 10.3390/biomedicines10010154

**Published:** 2022-01-12

**Authors:** Shofiul Azam, Ju-Young Park, In-Su Kim, Dong-Kug Choi

**Affiliations:** 1Department of Applied Life Sciences, Graduate School, BK21 Program, Konkuk University, Chungju 27478, Korea; shofiul_azam@hotmail.com; 2Department of Molecular Science and Technology, Ajou University, Suwon 16499, Korea; pink1209juyoung@empas.com; 3Department of Biotechnology, Research Institute of Inflammatory Disease (RID), College of Biomedical and Health Science, Konkuk University, Chungju 27478, Korea

**Keywords:** piperine, biosynthesis, metabolites, Alzheimer’s disease, Parkinson’s disease

## Abstract

Piperine (PIP) is an active alkaloid of black and long peppers. An increasing amount of evidence is suggesting that PIP and its metabolite’s could be a potential therapeutic to intervene different disease conditions including chronic inflammation, cardiac and hepatic diseases, neurodegenerative diseases, and cancer. In addition, the omnipresence of PIP in food and beverages made this compound an important investigational material. It has now become essential to understand PIP pharmacology and toxicology to determine its merits and demerits, especially its effect on the central nervous system (CNS). Although several earlier reports documented that PIP has poor pharmacokinetic properties, such as absorption, bioavailability, and blood–brain barrier permeability. However, its interaction with metabolic enzyme cytochrome P450 superfamily and competitive hydrophobic interaction at *Monoamine oxide B* (MAO-B) active site have made PIP both a xenobiotics bioenhancer and a potential MAO-B inhibitor. Moreover, recent advancements in pharmaceutical technology have overcome several of PIP’s limitations, including bioavailability and blood–brain barrier permeability, even at low doses. Contrarily, the structure activity relationship (SAR) study of PIP suggesting that its several metabolites are reactive and plausibly responsible for acute toxicity or have pharmacological potentiality. Considering the importance of PIP and its metabolites as an emerging drug target, this study aims to combine the current knowledge of PIP pharmacology and biochemistry with neurodegenerative and neurological disease therapy.

## 1. Introduction

Piperine (PIP), an alkaloid, omnipresent in foods/beverages, is currently one of the compounds of interest for showing numerous pharmacological benefits. PIP is a yellow crystalline solid compound mainly isolated from black pepper (*Piper nigrum*), but also found in other Piper species (Table 1). This compound is bio-synthesized from L-lysine and cinnamoyl-CoA precursor as a secondary metabolite (Figure 1). After a series of reactions, the bioactive PIP is yielded from piperonyl-CoA [1]. Although earlier studies re-ported that PIP has poor bioavailability [2] owing to its lipophilic nature, it can inhibit metabolic enzymes like CYP3A4 and drug transporter P-glycoprotein and modulate xenobiotics metabolism in human liver [3]. In the process, PIP inhibits the metabolism and efflux of xenobiotics, making PIP a potential bioavailability enhancer of drugs and nutraceuticals. For example, carbamazepine, a widely used anti-epileptic drug, has a very narrow therapeutic index, because carbamazepine is primarily metabolized by CYP3A4 and that yields carbamazepine-10,11-epoxide (CBZE) intermediate [4]. The CBZE controls both carbamazepine-mediated anti-seizure and toxicity [5], but the therapeutic index is so narrow that regular monitoring is essential to avoid toxicity. However, co-treatment with PIP has increased the carbamazepine area under the curve (AUC) and half-life in epileptic patients, thus enhancing the therapeutic window for carbamazepine in patients who require long-term therapy [6,7,8]. In addition, several studies recently developed a PIP-loaded nano-formulation, microemulsion, and intranasal delivery system to overcome poor bioavailability. Nevertheless, PIP treatment showed reduced depressive-like [9] and Huntington (HD)-like symptoms [10], showed improved cognitive function in an Alzheimer’s (AD) model [11], and halted neuronal cell death in a Parkinson’s (PD) model [12]. Altogether, the bioavailability enhancing property and the other pharmacological activities of PIP, and its metabolites have made them a hot target compound in recent research. Given the importance of these facts, several recent reviews have documented PIP and its metabolite’s different therapeutic aspects [13,14]; pharmacokinetic properties and medicinal chemistry [1]; and different pharmacological properties like anti-diabetic [15], anti-cancer [16,17], and autophagy regulation [18], as well as provided recent updates in PIP chemistry and new formulations [1,19]. Synthesizing this knowledge and present research context, the current review has attempted to summarize PIP effects on the prevention/treatment of different neurological and psychiatric diseases, with an emphasis on its pharmacokinetic properties.

## 2. Pharmacokinetic and Pharmacodynamic Properties of PIP

Knowledge of pharmacotherapy parameters, pharmacokinetic and pharmacodynamic, is imperative for drug development and dosing. The pharmacokinetic property of PIP has been analyzed on several occasions. In 1986, Bhat and Chandrasekhara [7] for the first-time showed, irrespective of the route of administration (170 mg/kg b.w. orally and 85 mg/kg b.w. intraperitoneally), that 97% of PIP could be absorbed in rats. That study also showed that a maximum amount of PIP could be detected in the liver from 1 h to 24 h of treatment and that the peak concentration could be found after 3 h of treatment. Whereas a trace amount of PIP could be detected in the blood serum and kidney at that time, but no trace of PIP can be found after 24 h. The major route of PIP excretion is the urinary system; irrespective of route of administration, PIP could be excreted in conjugation with glucuronides, sulphates, and phenols for up to 8 days, with a maximum rate at 1–4 days [7]. Another study reported that, after an oral gavaging of 170 mg/kg, the PIP concentration in the intestine gradually increased until 6 h and declined afterwards, and 96% of administered PIP was absorbed at that time [8], which was incomparably higher than its structural counterpart curcumin (63.5%) at the same time course.

Post-absorption from the intestine, PIP remains unchanged and has been found to be distributed to other organs like the liver, heart, spleen, lungs, and kidneys, with the highest concentration (50%) in the liver [9]. Moreover, the hydrophobic property of PIP allowed it to cross the intestinal nonpolar molecular barrier, which could be evident from PIP’s absorption rate, longer half-life, and apparent permeability coefficient [8]. Nevertheless, the hydrophobic nature of PIP also helping this compound to interact with human serum albumin, which assists PIP in transportation under physiological conditions [10].

To validate the major excretion route of PIP metabolites, both urine and feces were collected from PIP-treated rats, where no trace of PIP metabolites was detected in feces [11], suggesting that the kidney could the major route of excretion. PIP undergoes conjugation, glucuronidation, oxidation, and sulfonation, and yields metabolites including vanillin, piperonylic acid, piperonal, and piperic acid, which could be found in urine after 48–96 h of treatment.

Bioavailability of PIP is a major concern because of its hydrophobic nature; however, recent studies indicate that nano-formulation of PIP could enhance oral bioavailability [12,13]. The nanosized PIP not only enhanced bioavailability 2.7–3.65-fold, but also increased the dissolution rate by 3-fold and improved the postdosing brain concentration compared with non-formulated PIP [12]. Nevertheless, the dynamic property of PIP of forming a nonpolar complex with drugs or herbs promotes their permeability while modulating membrane dynamics via easy partitioning, indicating that PIP could modulate drugs or herb’s pharmacokinetic properties. For instance, co-treatment of PIP and curcumin increases the curcumin half-life and AUC by roughly twofold or more [8].

PIP consumption has been reported to inhibit various metabolic enzymes including CYP3A4, UDP-glucronyl transferase, UDP-glucose dehydrogenase, cytochrome BS, and aryl hydrocarbon hydroxylase. The pharmacokinetics of PIP could enhance the bioavailability of several other drugs/compounds when they are co-treated with PIP, including ciprofloxacin, resveratrol, carbamazepine, metronidazole, silybin, oxytetracycline, and phenytoin [10,14,15]. In addition, PIP co-treatment increases the anti-inflammatory efficiency and membrane permeability of resveratrol [16] and improves the serum half-life of coenzyme Q10 and β-carotene [17,18]. In summary, PIP not only provides a health benefit, but also its herb–drug or herb–herb interaction ability facilitates other pharmaceuticals’/nutraceuticals’ pharmacokinetic and dynamic properties. Moreover, using advanced technology like nano-formulations, PIP‘s bioavailability, absorption rate, transportation, and brain concentration after oral administration could be improved.

### 2.1. PIP Interaction with Cytochrome P450 Superfamily Enzymes

Cytochrome P450 (CYP) are the major enzymatic system responsible for many pharmaceuticals or nutraceutical’s metabolism. Inhibition of CYP enzymes might cause serious consequences, especially for those with a narrow therapeutic window. CYP3A is the major member of CYP subfamily that is extensively distributed in intestine epithelial cells and hepatocytes. There are two distinct ways of CYP inhibition; firstly, the parent compound could inhibit CYP directly and, secondly, the metabolites could inhibit CYP by a metabolism-dependent inhibition (MDI) pathway [19]. Food or food supplement ingestion could express or inhibit CYP activities, which makes it important to understand the possible drug–food interactions. PIP has been found to bind at the active site of CYP3A and inhibits enzymatic activities [20]. Cui et al. (2019) [20] showed that PIP moderately inactivates host CYP3A4 enzyme via the MDI pathway. They also reported that both carbene and *ortho*-quinone intermediates of PIP metabolites form a noncovalent bond at the active site of CYP3A4, resulting in enzyme inactivation. PIP contains a methylenedioxyphenyl (MDP) ring, which is metabolized by CYP and is possibly responsible for CYP3A inactivation; MDP metabolism yields carbene intermediates that form a carbene–heme–iron–porphyrin complex [21] and are suspected to cause host CYP3A inhibition. Thus, it is understood that PIP biotransformation is required for irreversible inactivation of CYP3A, and carbene intermediates are positively involved in this process.

### 2.2. PIP-Mediated Regulation of Drug Pharmacology

P-glycoprotein is a transmembrane efflux pump that functions as a drug transporter. P-glycoprotein effluxes drugs and/or xenobiotics from the inside to the outside of cells, which are lastly metabolized by CYP3A4. It has been suggested that p-glycoprotein inhibition could enhance or restore drug efficacy [22]. In addition, several reports showed that high lipophilicity of a compound is the most important physicochemical property for it to be a potential p-glycoprotein inhibitor [23,24]. Interestingly, replacement with 6,7-dimethoxytetrahydroisoquinoline, a moiety at the piperidine ring site, has shown improved inhibition of p-glycoprotein [22] and restored drug efficacy. Co-treatment of PIP (20 mg) once daily for 10 days in epileptic patients receiving carbamazepine (200 mg) has shown an enhanced effect of carbamazepine owing to a reduced level of hepatic p-glycoprotein and CYP3A4 [25]. This suggests that PIP is a potential target to modulate drug trafficking and ameliorate drug resistance.

### 2.3. PIP Effect on Other Drug Metabolizing Enzyme

PIP potentially inhibits UDP-glucose dehydrogenase enzyme—an enzyme that catalyzes glucuronidation and facilitates the elimination of xenobiotics—by a conjugated double bond and inhibits intestinal glucuronidation better than the liver [26]. Moreover, PIP inhibits liver and intestinal UDP-glucuronyl transferase and aryl hydrocarbon hydroxylase enzyme [27]. Oral treatment of PIP has been shown to inhibit UDP-glucuronyl transferase and aryl hydrocarbon hydroxylase enzyme activities in rat liver and prolong hexobarbital-induced sleeping time and zoxazolamine-induced paralysis time [27], suggesting that PIP is a potential inhibitor of drug metabolism.

## 3. PIP Metabolism and Effect on Neurodegeneration 

### 3.1. Catabolism of PIP

Examining PIP-incubated hepatocytes (of rats, dogs, mouse, and human), it was revealed that at least 20 primary and secondary metabolites are formed from PIP [28]. PIP is fragmented (the basic fragmentation is shown in Figure 2) by several steps of reaction to be converted into these metabolites, generally starting with the loss of the piperidine ring followed by the cleavage of the amide bond. Firstly, PIP undergoes oxidation to form catechol derivatives, then passes through methylation, glucuronidation, glutathione conjugation, and hydroxylation to open the piperidine ring. Then, the hydroxylation intermediate carbinolamine derivatives are formed and the piperidine ring is opened, which produces alcohol and acid derivatives. Finally, stable hydroxylated or decarboxylated metabolites are also yielded by the hydroxylation or decarboxylation process.

### 3.2. Excretion of Metabolites

Studies are suggesting that the kidney is the major route of excretion for post-metabolized PIP [11]. Oral gavaging of PIP in rats yielded a number of metabolites by oxidation and/or cleavage of the piperidine ring, which were not found in feces, but only in the urine sample (Figure 2). However, it is to be noted that PIP metabolites from rats and human are not the same; thereby, Li et al. (2020) have used mouse, human, and rat hepatocyte to identify, characterize, and distinguish PIP metabolites of different species [28]. Using a thorough screening and analytical techniques containing LC/DAD-HRMS and MS/MS, they have identified 20 metabolites; of them, 5 are primarily from PIP and the remaining 15 are secondary to those 5 primary metabolites (C11, 12, 15, 16, and 18; shown in Figure 2). Additionally, all isolated metabolites from different species are thus far not the same, except nine (Table 2). On the other hand, until now, PIP metabolites of different species including human, dog, mouse, or rat have been reported in several studies, but at the very least, we know about these metabolite’s pharmacology or toxicology. It has been suggested that the metabolites found in rat urine are plausibly due to the instability of the primary biotransformation or could be due to the enzymatic interaction at the renal system [11,29]. However, a recent study has found that the presence of human liver microsomes might play a role in changing the structural conformations of PIP, which could produce multiple stable and reactive metabolites of PIP [30].

Knowledge of PIP metabolites is important as it does not excrete as it occurs after oral ingestion and because of its omnipresence in food/beverage. Thus, further study should be done to thoroughly assess the pharmacological/toxicological effects of this alkaloid.

### 3.3. SAR of PIP in Neuropharmacology

Several metabolites from PIP synthesized by the derivatization of amide function, hydrogenation of the aliphatic sidechain, replacing amide with an ester group, carbonyl group removal, and modifying the MDP ring [31]. These metabolites possess selective or non-selective action over monoamine oxidase (MAO) and adenosine 2A (A2A) receptor. MAO is an enzyme that regulates the inactivation pathways of several neurotransmitters including noradrenaline, adrenaline, dopamine, and 5-hydroxytryptamine. MAO have two isoforms, -A and -B, and selective inhibition of MAO-B offers a therapeutic benefit in AD and PD treatment, while MAO-A inhibition provides benefits in depression and anxiety. Rigorous clinical and pharmacological analysis suggests that selective inhibition of this enzyme plausibly slows and/or reverses neurodegenerative progression in PD and AD. The structure–activity relationship (SAR) of PIP derivatives that formed by replacing piperidine ring with the propyl, butyl, diethy, *N*-methyl phenyl, methoxy, and ethoxyl group showed non-selective inhibition of both MAO-A and -B (Figure 3) [32]. Substitution of piperidine with small molecule amines improved selectivity to MAO-B, but showed poor affection to MAO-A. This indicates that piperidine moiety plays a major role in PIP-mediated neuropharmacology. Nevertheless, catalytic hydrogenation and/or carbonyl group removal of PIP could impair its MAO-A and -B enzyme inhibitory activity (Figure 3). Interestingly, substitution of MDP moiety by phenolic hydroxylation increased MAO-A selectivity, but reduced affection to MAO-B [32].

Adenosine, a nucleoside formed by degradation of ATP, is located in both glial and neuronal cells and acts as a neuromodulator. One major member of this receptor family is A2A, which forms a heteroreceptor complex with metabotropic glutamate receptor 5 (mGluR5) and dopamine 2 receptor (D2R). These heteroreceptor complexes may increase protein aggregation and toxicity and exaggerate neurodegeneration [33]. Thus, antagonists of A2A receptor alleviate PD symptoms; however, A2A antagonists are also a potent MAO-B inhibitor. For example, the oxidation or phenyl ring substitution of the substrate (*E*,*E*)-8-(4-phenybutadien-1-yl)caffeinyl significantly improved MAO-B selectivity as well as A2A antagonism [31,34]. In contrast, modification of caffeinyl ring with ethyl derivatives showed weak affinity to MAO-B, but had positive affinity to A2A receptors [31,34]. Again, this suggests that modulation of the piperidine ring could modulate A2A receptor activities.

Consistent with Figure 3, a recent study has synthesized PIP-based derivatives by hydrolyzing PIP to piperic acid and analyzed their activity of MAO-A and MAO-B inhibition by a continuous fluorometric assay combined with computational docking [35]. The author showed that the replacement of the piperidine ring with a bulk aromatic ring increased the specificity and affinity of MAO-A inhibition owing to the presence of a large hydrophobic substrate cavity in MAO-A. Contrarily, replacement of the piperidine ring with a small amine moiety improves MAO-B selectivity and inhibition rather than MAO-A. Moreover, several apolar loops exist at the proximity of the C-terminal helix of MAO-B that facilitate additional membrane-binding via hydrophobic interactions and control opening and/or closing of the active site [36]. On the other hand, the MDP moiety showed affinity to form a side chain with polar residues like Thr314, Thr201, Ser200, and Thr202 at the MAO-B active site, and that hydrophobic interaction selectively increases MAO-B inhibition [35].

## 4. PIP in Neurological and Psychiatric Diseases

### 4.1. AD

Oxidative stress plays a vital role in the pathogenesis of AD and subsequent cognitive impairment. A recent study used intracerebroventricular (ICV) injection of streptozotocin (STZ) for AD modeling in rats by inducing severe oxidative stress and impairing hippocampal neurotransmission [37]. In their model, a low dose of PIP (2.5 mg/kg/day for 4 weeks) improved the memory task and long-term potentiation in animals. Interestingly, chronic PIP administration resulted in more efficient synaptic plasticity than donepezil. Chronic treatment of PIP for 23 days also improved locomotor activity and neurotransmission by reducing oxidative-nitrosative stress in cerebrospinal fluid and hippocampus and enhanced cholinergic function in ICV-STZ infusion-induced sporadic AD mouse model [38,39]. Moreover, PIP treatment inhibits glutamate release from hippocampal nerve endings by reducing Ca^2+^ current in rats [40], which could increase paired-pulse facilitation and, subsequently, increase short-term synaptic plasticity. Acetylcholine (ACh) receptor is abundantly expressed in the hippocampus and regulates plasticity and cognitive function. PIP and its derivatives have shown potential inhibition of ACh esterase (AChE) [41], an enzyme that catalyzes and breaks ACh into acetate and choline. Although the molecular basis of ICV-STZ infusion-induced memory deficits is unknown, STZ infusion causes a sharp increase in amyloid-β (Aβ), synapsin, and tau-phosphorylation [42]. Therefore, it could be presumed that PIP treatment might modulate misfolded Aβ and tau hyperphosphorylations. A study showed PIP treatment not only benefits cholinergic functions, but also benefits from Aβ and tau pathogenesis by inhibiting β-secretase [43,44].

Further, as we discussed earlier, PIP is a potential bioenhancer of drugs and/or nutraceuticals. The combination of existing and/or emerging therapeutics with PIP has become popular practice. With such intentions, Head et al. [45] tested a medical cocktail containing PIP (epigallocatechingallate 36.3% by wt., PIP 3.0% by wt., N-acetyl-l-cysteine 15.3% by wt., curcumin 36.3% by wt., and R-lipoic acid 9.1% by wt.) in aged dogs in an AD model. After 3 months of treatment, the dogs showed improved spatial attention and reduced cognitive impairment compared with non-treated age matching dogs, though brain and CSF Aβ remains unchanged. As that study observed neither Aβ plaque accumulation nor soluble and/or insoluble Aβ1-40 and Aβ1-42, it could draw few conclusions. For example, the presence of PIP in the cocktail not only improves other molecules’ pharmacokinetic property, but also its potential antioxidant property ameliorating cognitive impairment independently of Aβ or PIP could be inhibiting β-secretase [43,44] to prevent pathogenic Aβ fragmentation and accumulation.

One major limitation of PIP treatment is its poor bioavailability, for which PIP requires a high dose to reach maximum efficiency. In addition to overcoming this limitation, nano-formulation or formulations containing PIP nanoparticle are being developed. Intranasal treatment of PIP-loaded chitosan nanoparticles showed 20-fold better pharmacokinetics than oral treatment in male AD rats [46]. PIP solid lipid nano-formulation also improved blood–brain barrier permeability and ameliorated mobility in AD animals by reducing acetylcholinesterase activity and Aβ plaque and tangles [47]. The PIP loading into nonionic surfactant-based ME formulations could increase its low-density lipoprotein receptor affection and might increase blood–brain barrier permeability [48]. This might enhance its potentiality in AD treatment. However, the surfactants’ concentration must be carefully adjusted, as a high concentration might cause nephrotoxicity [49].

### 4.2. PD

Dopamine depletion is a major biomarker of PD pathogenesis. MAO-B is an important catalyst of dopamine synthesis, and the inhibition of MAO-B might deplete dopamine synthesis. MAO-B is a flavin adenine dinucleotide (FAD) containing enzyme, located in the mitochondrial outer membrane [50]. We discussed above that PIP and its metabolites are a potential inhibitor of both MAO-A and -B. The presence of the pentacyclic ring in the structure could reduce the MAO-B inhibition potential of PIP, while removal of nitrogen from the pyrrolidine ring might increase MAO-B inhibition selectively in the PD model [51]. Moreover, PIP treatment has diminished lipid peroxidation and increased glutathione in 6-hydroxydopamine (6-OHDA) lesioned rat’s striatum [52]. PIP treatment also induced atuophagolysosome via Akt/mTOR pathway activation and reduced neuronal degeneration in substantia nigra in rotenone-injected PD mice [53]. A PIP-mediated increase in protein phosphatase 2A (PP2A) ameliorated mitochondrial injury, mitochondrial membrane permeability transition pore (mPTP) impairment, and oxidative stress [53,54]. In addition, SAR of PIP showed substitution of the aromatic group with a heterocycle and piperidine with diethylamine and ethylamine could enhance its pharmacokinetics and blood–brain barrier permeability. Meanwhile, such a modification also improved PIP derivatives’ efficiency in Nrf2/Keap1 pathway activation, thus providing neuroprotection in the PD model [55]. PIP gavaging (25–100 mg/kg b.w.) could restore purinergic receptor P2X ligand-gated ion channel 4 (P2RX4) activity and promote autophagic flux in the α-synuclein overexpressed mouse PD model [56]. Autophagy plays an important role in PD, while α-synuclein impairs autophagic flux by inhibiting syntaxin 17 (STX17), which promotes autophagolysosome membrane fusion by P2RX4 inactivation. PIP treatment re-activated P2RX4, promoted autophagic flux, improved neuronal cell viability, and ameliorated motor impairment. PIP (10–40 mg/kg) treatment for 40 days in a carotid artery occlusion model of PD dementia showed reduced expression of mammalian target of rapamycin (mTOR) regulator miR-99a-5p and pro-inflammatory factors (IL-6 and TNF-α), as well as slowing PD dementia pathology [57].

### 4.3. Huntington’s Disease

HD is a complex neurodegenerative disease characterized by progressive motor dysfunction and mental abnormality owing to progressive loss of striatal neurons. PIP activity in an HD model has not been extensively researched. A recent study used 3-Nitropropionic acid (3-NP), a mycotoxin, to induce an HD-like model in rats and test PIP potentiality. PIP treatment restored ATP production and ameliorated striatal degeneration in basal ganglia and motor function in HD rats [58].

### 4.4. Epilepsy

PIP has been studied by several research groups for the therapeutic potential in epilepsy, a neurological disease characterized by frequent seizures. PIP received special attention in anti-epileptic treatment for its potential MAO inhibitory and neuroprotective activity. In temporal lobe epilepsy or post status epilepticus (pSE) animal model, PIP treatment (25 mg/kg for 10 days) restored the serotonin level and modulated MAO and γ-amino butyric acid (GABA)ergic pathway [59]. It is also reported that PIP treatment increased GABA, glycine, and taurine, and reduced nitrite in the sera and brain of a pilocarpine-induced epileptic mouse model [60]. In a maximal electroshock (MES)-induced seizure model of epilepsy, PIP treatment (10 mg/kg) reduced the morbidity by negatively regulating the Na^+^ channel, which delayed the onset of tonic clonic seizures [61], indicating that PIP could modulate the Na^+^ channel to reduce tonic clonic convulsion and related morbidity in epilepsy. For instance, blocking of transient receptor potential cation channel subfamily V member 1 (TRPV1) by a selective antagonist suppressed PIP-mediated anti-convulsant activity [62], indicating that TRPV1 might also be involved in the PIP mechanism. Unlike capsaicin, which forms a hydrogen bond at T551 and E571 ligand-binding sites of TRPV1, PIP might directly interact with the pore-forming S6 segment to induce channel opening [63]. Besides, the use of PIP as a sub-therapeutic agent or therapy enhancer revealed another chapter for PIP importance. Co-treatment of PIP and carbamazepine in epilepsy patients showed a significant increase in carbamazepine bioavailability, absorption, and AUC and reduced elimination half-life [15,64].

### 4.5. Other Neurological Diseases

PIP has also gained widespread attention in a few other neurological disorders. For example, a recent study showed that PIP interacts with r(CGG)^exp^ RNA motifs and that this interaction improves r(CGC)^exp^ associated splicing defects and RNA translation in fragile X-associated tremor/ataxia syndrome (FXTAS) [65]. FXTAS is a neurological disease where the trinucleotide CGG repeats more than 200 times and results in clinical features like multisystemic atrophy, intention tremor, cerebellar and gait ataxia, and dementia. The interaction of PIP with r(CGG)^exp^ in an FXTAS model indicates its therapeutic importance in FXTAS treatment. Moreover, PIP could modulate the serotonin (5-HT) level in the brain and function as an anti-depressant and could synergize antidepressant activity of tetralin (5-HT_1A_ agonist) and anpirtoline (5-HT_1B_ agonist) [66]. Interestingly, PIP-mediated anti-depressant activity involves BDNF/TrkB signaling cascade as well, and chronic treatment of PIP could increase BDNF protein concentration in the hippocampus and cortex of the mouse model of depression [67]. Moreover, a PIP derivative, SCT-66, has been reported to modulate GABA_A_ and TRPV1 receptors more effectively than PIP [68]. SCT-66 has also been reported to have much better anxiolytic than PIP, without changing body temperature.

Neuroinflammation is the most common pathophysiological feature of many neurodegenerative diseases including AD and PD. ROS production and glial cells’ activation is the key to the neuroinflammatory response. A series of pro-inflammatory cytokines including tumor necrosis factor α (TNF-α), interleukin 1β (IL-1β), and interleukin 6 (IL-6) are released and exaggerate the neurodegeneration. Among the other pathways, nuclear factor erythroid 2 related factor 2 (Nrf2) and heme oxygenase-1 (HO-1) are critical regulator of endogenous oxidative stress (Figure 4). PIP could activate Nrf2 signaling and downregulate NF-κB activation [69], a nuclear factor that could translate an inflammatory cytokine like IL-1β. PIP treatment (10 mg/kg, once daily) for 15 days has been shown to ameliorate behavioral deficits and pro-inflammatory cytokines (IL-1β, IL-6, and TNF-α) by reducing infarct volume and NF-κB activation in a middle cerebral artery occlusion (MCAO) rat model [70].

Depression is one of the major psychiatric disorders featured by fatigue, pain, cognitive dysfunction, depressed mood, and/sleep disturbances. PIP (5–20 mg/kg) treatment for 4 weeks showed anti-depressant activities in mildly stressed rats, plausibly by enhancing the serotonin level at the cerebral cortex and limbic area [71]. Besides, chronic administration of PIP and its derivative antiepilepsirine (10–20 mg/kg) for 2 weeks significantly increased the dopamine and serotonin level in the hypothalamus and hippocampus, resulting in anti-depressant-like activities [72]. Another study showed that pre-treatment of PIP (5–20 mg/kg, i.p.) in a stress-induced anxiety model significantly reduced the nitrite level and suppressed anxiety-like activities [73]. 

## 5. Therapeutic Index and Future Perspective

Indeed, PIP is an important natural compound that has multiple health benefits. Yet, there are a few limitations halting its wide application. Firstly, the hydrophobicity, which caused its poor absorption and bioavailability. Another major concern of PIP is toxicity due to the presence of the MDP group in its structure [11], which is common in carcinogenic compounds like safrole. Though the carcinogenicity of PIP has not been confirmed yet, treatment of 2 mg/kg on 3 days of a week for 3 months caused tumors in mice [75], showed cytotoxicity to cultured neurons of rat brain [76], and showed the formation of benzo[a]pyrene-DNA adducts that caused lung fibroblast cytotoxicity [77]. Consistent oral administration of PIP at 1.12–4.5 mg/kg body weight for 5 days in mice reduced leukocytes and myogenic response of B lymphocytes and increased neutrophils [78]; however, the lowest dose (1.12 mg/kg) had no immunotoxicity. Moreover, no adverse effects level (NOAEL) of PIP have been observed at 5 mg/kg/day consumption in humans [79], which indicates that PIP consumption through different foods has no toxic effect other than irritation if a higher amount than normal is consumed. Besides, subacute toxicity testing of PIP oral and intravenous dosing showed that up to 100 mg/kg of oral dosing is nontoxic, whereas intravenous dosing has an LD_50_ of 15.1 mg/kg body weight of adult mice [10]. As PIP is mainly administered via the oral route, these data suggest that PIP is nontoxic and safe, and might cause irritation if overdosed.

Apart from a few adverse effects reported, so far, for overdosing of PIP, the main concern is high lipophilicity. Lipophilicity is an important physicochemical property of a compound that has parabolic relationship with in vivo brain penetration. High lipophilic compounds often experience low blood–brain barrier access owing to increased non-specific plasma protein bindings and vulnerability to P450 metabolism that led to rapid clearance of compounds. Because of this high lipophilic property, a low concentration of PIP is observed in the brain, limiting its neuroprotective effect at a low dose. Several recent studies have addressed this issue and resolved it on their own. For example, tween-modified monoolein cubosomes (T-cubs) loaded with PIP showed higher efficacy over free drug without affecting PIP’s cognitive function restoration effect in an AD model [80]. Microemulsification of PIP also improved its bioavailability and blood–brain barrier permeability significantly [48]. Similarly, intranasal delivery of PIP-loaded chitosan nanoparticles [46] or PIP-SLNs’ formulation via emulsification coated with PS80 [47] showed that a lower dose of PIP could be effective as well. However, as PIP is a potential bioenhancer, it could be administered with other drugs or herbs to potentiate their therapeutic efficacy in neurodegenerative diseases, like with curcumin, resveratrol, quercetin, carbamazepine, metronidazole, silybin, oxytetracycline, and/or phenytoin [10,81].

PIP is a potential antioxidant compound as well as an important therapeutic candidate for neurodegenerative diseases. In this current review, although we discussed beneficial sites of PIP, PIP is also reported for toxicity at a high dose. Extensive structural and functional chemistry-based research is needed to overcome this limitation. Besides, few studies until now have reported on PIP pharmacology on neurodegenerative diseases, especially of human trials. Thus, the answers to a few questions remain unclear, such as does PIP have role over misfolded protein clearance (Aβ, tau or αS)? Is PIP treatment protecting BBB integrity from environmental toxin-induced condition? A lack of studies on PIP in neurodegenerative disease-like models have limited our study from explaining more details about the therapeutic application of PIP. However, the strong point of this review is that we now know how to enhance PIP bioavailability, and that PIP co-treatment enhances other therapeutics’ bioavailability.

## Figures and Tables

**Figure 1 biomedicines-10-00154-f001:**
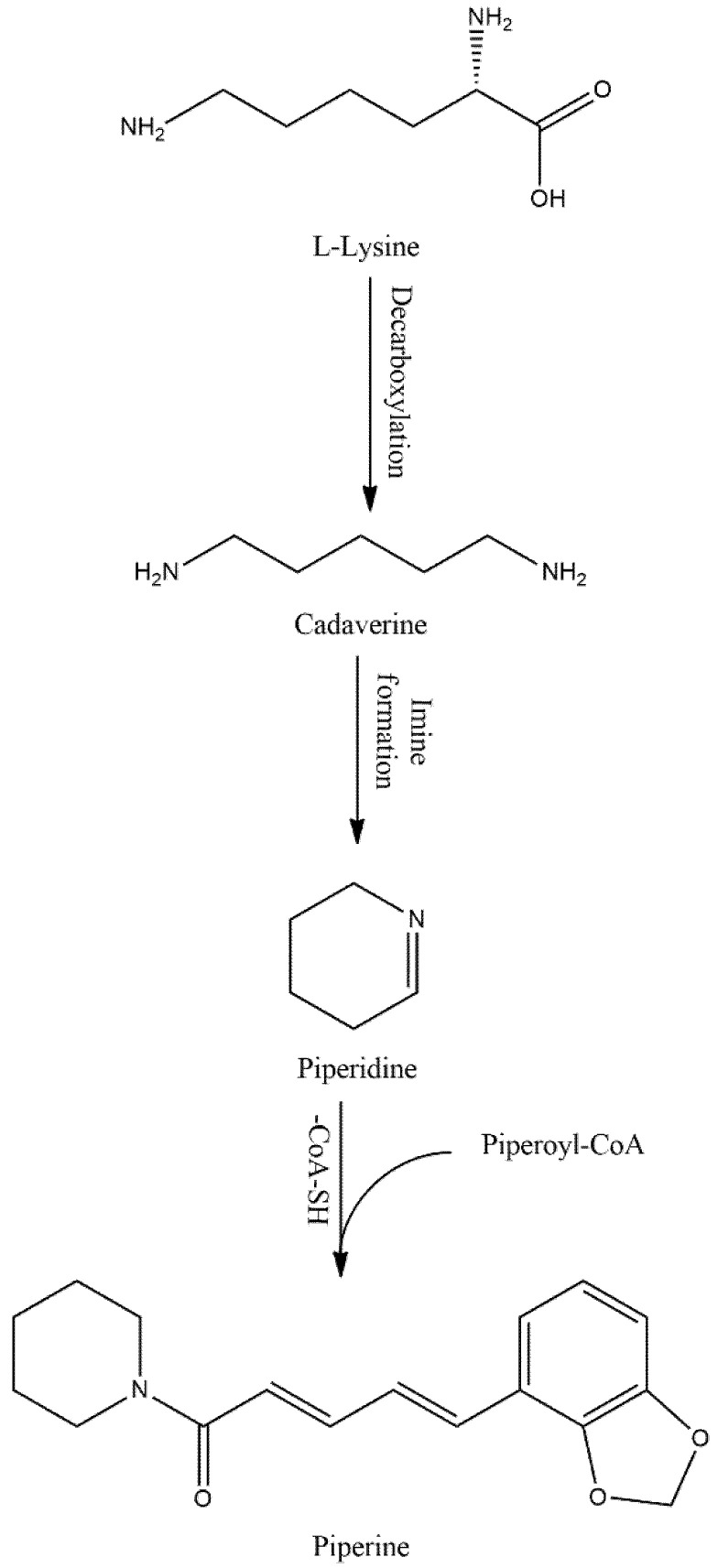
Possible biosynthesis pathway of PIP.

**Figure 2 biomedicines-10-00154-f002:**
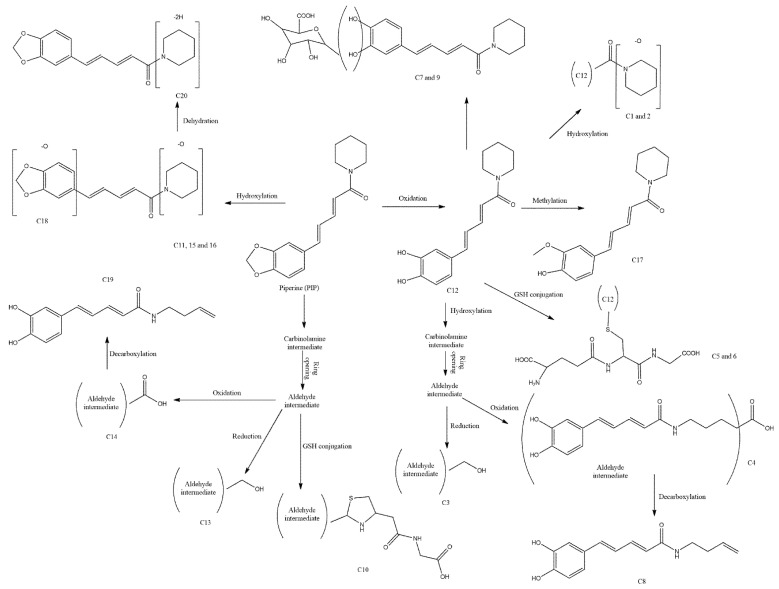
Possible biotransformation of PIP metabolites in hepatocyte. Based on Li Y et al. [28], 20 metabolites could be detected from PIP in the liver. Li and colleagues used hepatocytes of four different species (human, rat, mouse, and dog) and found that a few of these 20 metabolites could be detected in all species (C6, C9, C11, C12, C13, C14, C15, C19, and C20), but not all. C1–5, C7, and C17 were found to be present in both mouse and rat species, while C10, C16, and C18 were exclusively present in mouse. Metabolite C8 was observed only in rat hepatocyte. Other than common metabolites for all species, C7 in human and C2–5 in dog hepatocytes were observed.

**Figure 3 biomedicines-10-00154-f003:**
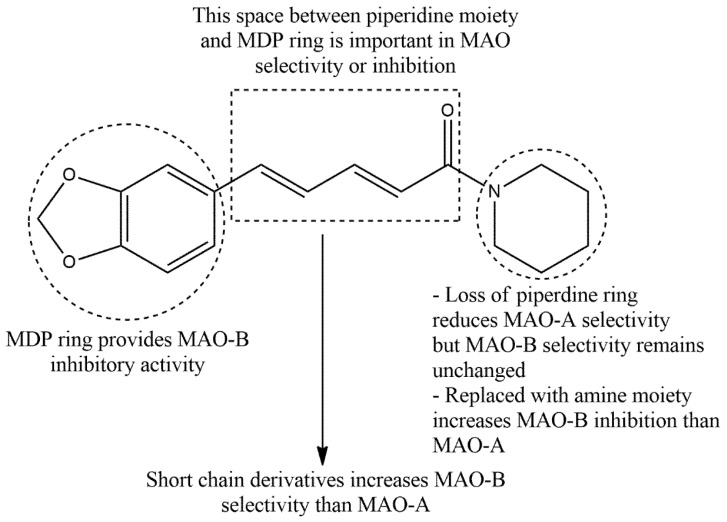
The structure–activity relationship of PIP in MAO selectivity.

**Figure 4 biomedicines-10-00154-f004:**
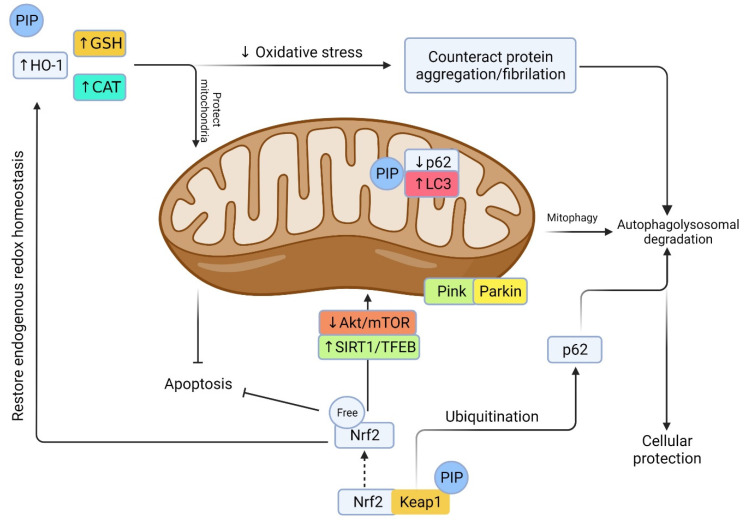
Prospective machinery of PIP in oxidative stress-induced neurodegenerative events. Oxidative stress arises from a decrease in endogenous antioxidant defense including HO-1, GSH, CAT, and Nrf2. This leads to a cascade of events including protein oxidation/glycation/mitochondrial damage, protein aggregation, and impaired autophagy/lysosome vacuoles. PIP is a potential antioxidant that could induce oxidative stress via promoting antioxidant defense, reducing ROS, and preserving mitochondrial integrity. On the other hand, Keap1 degradation and Nrf2 activation by PIP could counteract oxidative stress, reduce mitochondrial damage, promote autophagy by degrading/ubiquitination of p62/sequesterosome, promote autophagic flux by converting microtubule-associated protein 1A/1B-light chain 3 (LC3), promote mitophagy and mitobiogenesis via distinct pathway, and rescue from cellular apoptosis and degeneration [74].

**Table 1 biomedicines-10-00154-t001:** Estimated amounts of PIP in different plant parts.

Sources	Plant Parts	Estimated Amount (%)
*Piper nigrum* [2]	Fruit	1.7–7.4
*Piper longum* [3]	Spike and root	5.9
	Fruit	0.03
*Piper chaba* [3,4]	Fruit	0.95–1.32
*Piper guineense* [5]	Fruit	0.23–1.1
*Piper sarmentosum* [6]	Root	0.20
	Stem	1.59
	Leaf	0.104
	Fruit	2.75

Estimated amounts (%) were calculated on a dry wet basis.

**Table 2 biomedicines-10-00154-t002:** Variations of metabolite production in different species.

Metabolites	Mouse	Rat	Dog	Human
C1				
C2				
C3				
C4				
C5				
C6				
C7				
C8				
C9				
C10				
C11				
C12				
C13				
C14				
C15				
C16				
C17				
C18				
C19				
C20				

Color shades indicate the prevalence (green), presence (blue > yellow > orange), and absence (red) of metabolites across different species.

## Data Availability

Not Applicable.

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
