# Peer review of "Piperine and Its Metabolite’s Pharmacology in Neurodegenerative and Neurological Diseases"

_biomedicines, 2022, doi:10.3390/biomedicines10010154_

Round 1

Reviewer 1 Report

This manuscript provides a detailed review of the pharmacological and biochemical properties of piperine (PIP), an alkaloid compound commonly found in food and beverages, and explores its physiological relevance in the prevention or treatment of a broad spectrum of human diseases. The review further discusses the interactions of PIP with key molecular targets including cytochrome p450, P-glycoprotein, and UDP-glucose dehydrogenase enzyme.

- My main suggestion is that author expand the discussion and provide more details (perhaps complemented by a figure panel) on the recent advances that have helped overcome PIP’s limitations.

- There are minor typographical errors such as “but PIP can "inhibit interact” with meta-bolic enzymes” including the appearance of unwanted hyphens throughout the manuscript.

Author Response

This manuscript provides a detailed review of the pharmacological and biochemical properties of piperine (PIP), an alkaloid compound commonly found in food and beverages, and explores its physiological relevance in the prevention or treatment of a broad spectrum of human diseases. The review further discusses the interactions of PIP with key molecular targets including cytochrome p450, P-glycoprotein, and UDP-glucose dehydrogenase enzyme.

- My main suggestion is that author expand the discussion and provide more details (perhaps complemented by a figure panel) on the recent advances that have helped overcome PIP’s limitations.

Response: Thank you for this generous suggestion, few major concerns with PIP includes bioavailability, lipophilicity and affinity to CYP. We have briefly discussed about bioavailability enhancement and decreasing lipophilicity in our current article. On the other hand, PIP affinity to bind active site of CYP, although, decreasing PIP pharmacological activities, but could be useful to potentiate other therapeutics efficiency. In addition, some studies have shown that high dose of PIP has toxicity. We explained SAR modulation or adjusting dose regimen might reduce this toxicity in our study.

- There are minor typographical errors such as “but PIP can "inhibit interact” with meta-bolic enzymes” including the appearance of unwanted hyphens throughout the manuscript.

Response: We have corrected those typo errors in our revised version.

Reviewer 2 Report

  • R 14- Please rephrase this sentence” this alkaloid and its reactive metabolites could be a potential target”
  • R 18- ”PIP pharmacological and toxicology”- Please replace with ”PIP pharmacology and toxicology”
  • R22- ”mono amino oxide B (MAO-B)”- Please replace with ”monoamine oxidase B”
  • R 35- Piper nigrum- Piper nigrum
  • R 36 ”this compound bio-synthetically synthesized”-please replace with ”this compound bio- synthesized”
  • R40- inhibit interact-What does it mean?
  • R 63 - Which are the psychiatric diseases?
  • R 77 vesus R90- ”PIP does not excrete as it is through urine;” versus ” The major excretion route of PIP is kidney as no trace of PIP metabolite could be detected in feces”. Please change: The major excretion route of PIP metabolits …..
  • R 105 ” This pharmacodynamics” is not correct. Please change with ”This pharmacokinetics….”
  • R 117 Please change ”pharmaceuticals or nutraceuticals elimination with pharmaceuticals or nutraceuticals metabolism”
  • R 125-126-Please rephrase ” The study by Cui et al. (2019) [36], PIP moderately inactivates…”
  • R 133 Please change ”PIP effect on drug trafficking”
  • 175-176 Please change ”a study orally gavaged PIP to rats”
  • R 175-177-Please rephrase this sentence” As kidney is the major route of excretion for post-metabolized PIP, a study orally gavaged PIP to rats and showed that a number of PIP metabolites forming by oxidation  and/or by cleavage of piperidine ring are excreting through urine only, and metabolites were found in feces”
  • R 252. Please mention the duration for low dose administration
  • R 270 Please mention the composition of this cocktail
  • 291-292 Please rephrase ” The enzyme MAO-B is an important catalyst of dopamine metabolism, and that inhibition of MAO-B might in dopamine level.
  • R299 Please explain this abbreviation”6-OHDA”
  • R319-320 Please rephrase ” HD is a complex neurodegenerative disease characterized by motor dysfunction and mental abnormality due to progressive striatal neurons”
  • R 355 Please correct ” delayed tonic clonic seizures”
  • R401-402 Please rephrase ”PIP has limited blood-brain barrier concentration and that is limiting its neuroprotective effect at low dose”
  • R401-402 Explain why high lipophilicity affects the crossing of the blood-brain barrier?
  • Please give more data from human trials
  • Please mention the limitations of your study
  • Please check all the abbreviations
  • Please check the correctness of the English text

Author Response

  • R 14- Please rephrase this sentence” this alkaloid and its reactive metabolites could be a potential target”

Response: We have revised that sentence in our updated manuscript.

  • R 18- ”PIP pharmacological and toxicology”- Please replace with ”PIP pharmacology and toxicology”

Response: We have corrected the typo mistake.

  • R22- ”mono amino oxide B (MAO-B)”- Please replace with ”monoamine oxidase B”

Response: Thank you for this suggestion, we have corrected the typo mistake.

  • R 35- Piper nigrum- Piper nigrum

Response: We did italicize the name in revised version.

  • R 36 ”this compound bio-synthetically synthesized”-please replace with ”this compound bio- synthesized”

Response: Thank you for this correction, we have made correction.

  • R40- inhibit interact-What does it mean?

Response: We apologize for this typo mistake, corrected in revised manuscript.

  • R 63 - Which are the psychiatric diseases?

Response: Thank you for this correction suggestions. Most common psychiatric diseases include depression, anxiety, PTSD, schizophrenia and bipolar disorder. PIP is a potent anti-depressant and could ameliorate epilepsy-associated depressed mood disorder. We have added anti-depressant and anxiolytic activities of PIP in revised manuscript (Line: 375-383).

  • R 77 vesus R90- ”PIP does not excrete as it is through urine;” versus ” The major excretion route of PIP is kidney as no trace of PIP metabolite could be detected in feces”. Please change: The major excretion route of PIP metabolits …..

Response: We have modified both sentences in our revised version.

  • R 105 ” This pharmacodynamics” is not correct. Please change with ”This pharmacokinetics….”

Response: Thank you for this suggestion, we have corrected as per suggestion.

  • R 117 Please change ”pharmaceuticals or nutraceuticals elimination with pharmaceuticals or nutraceuticals metabolism”

Response: Thank you for this suggestion, we have modified as per suggestion.

  • R 125-126-Please rephrase ” The study by Cui et al. (2019) [36], PIP moderately inactivates…”

Response: We have rephrased that sentence in our revised manuscript.

  • R 133 Please change ”PIP effect on drug trafficking”

Response: We have changed the sub-section title to “PIP-mediated regulation of drug pharmacology”

  • 175-176 Please change ”a study orally gavaged PIP to rats”

Response: Appreciate this suggestion, we have modified the sentence.

  • R 175-177-Please rephrase this sentence” As kidney is the major route of excretion for post-metabolized PIP, a study orally gavaged PIP to rats and showed that a number of PIP metabolites forming by oxidation  and/or by cleavage of piperidine ring are excreting through urine only, and metabolites were found in feces”

Response: Appreciate this suggestion, we have modified the sentence.

  • R 252. Please mention the duration for low dose administration

Response: It was a 4 week long study if different doses of PIP, we have mentioned the duration in the manuscript.

  • R 270 Please mention the composition of this cocktail

Response: The medical cocktail compositions were 203 mg epigallocatechingallate or 36.3% by weight, piperine, 3.0% by weight, N-acetyl-l-cysteine 15.3% by weight, curcumin 36.3% by weight and R-lipoic acid 9.1% by weight. We have mentioned in the updated manuscript.

  • 291-292 Please rephrase ” The enzyme MAO-B is an important catalyst of dopamine metabolism, and that inhibition of MAO-B might in dopamine level.

Response: We have rephrased that sentence.

  • R299 Please explain this abbreviation”6-OHDA”

Response: We have elaborated the abbreviation “6-OHDA” in text and in “abbreviation” section.

  • R319-320 Please rephrase ” HD is a complex neurodegenerative disease characterized by motor dysfunction and mental abnormality due to progressive striatal neurons”

Response: Appreciate this suggestion, we have made the correction.

  • R 355 Please correct ” delayed tonic clonic seizures”

Response: Tonic clonic seizures or convulsion is a combined form of seizure that begins with muscles stiffening (called tonic phase), then starts next phase of rhythmic jerking of arms and legs (called clonic phase). We have rephrased that sentence in revised version.

  • R401-402 Please rephrase ”PIP has limited blood-brain barrier concentration and that is limiting its neuroprotective effect at low dose”

Response: Thank you for this suggestion, we have rephrased this sentence in our revised version.

  • R401-402 Explain why high lipophilicity affects the crossing of the blood-brain barrier?

Response: Lipophilicity is an important physicochemical property of a compound. That determines compounds absorption, distribution, metabolism, elimination, plasma protein affinity and BBB penetration ability. High lipophilic compounds often suffer from low brain concentration mostly because of non-specific plasma protein bindings and/or rapid clearance due to P450 metabolism. We have briefly explained how does lipophilic property of a compound affect BBB permeability in our revised manuscript.

  • Please give more data from human trials

Response: We have mentioned several studies that used human liver sample to assess PIP and its metabolites pharmacology. However, not so many human trials had been conducted, especially for neurodegenerative disease models, so we are unable to precisely review at this point. This is one of the limitations of this study.

  • Please mention the limitations of your study

Response: Thank you for this suggestion, we have now discussed our study limitations in “therapeutic index and future perspective” section.

  • Please check all the abbreviations

Response: Thank you for this suggestion, we have added “abbreviation” section after “conflict of interest” section. Also, we have explained all abbreviations on their first appearance in text.

  • Please check the correctness of the English text

Response: Appreciate this suggestion, we have rechecked whole manuscript thoroughly to enhance adherence-coherence and correctness of sentences.

Reviewer 3 Report

Overall and general recommendation

The concept of the study is not novel but the authors have tried to justify with the title. My major concern about this review is that the authors have discussed only four diseases out of which, three are neurodegenerative diseases. The authors have discussed only one neurological disease i.e., Epilepsy. So, the title is not fully satisfied by the content. My suggestion is that either the title of the manuscript should be modified or the content should include more neurological diseases.

Other major issues which needs attention are:

  1. The manuscript also has multiple grammatical mistakes which I have listed in the table below:

Serial no

Section/Location

Existing text

Comments/proposed text

1

Abstract, line 14

sug-gesting

suggesting

2

Abstract, line 14

target

should be replaced with ‘intervention’

3

Abstract, line 16

neuro-degenarative

neurodegenarative

4

Abstract, line 17

compound important study material

 Should be replaced with ‘important investigational compound’

5

Abstract, line 18,19

 effect in central nervous system (CNS)

Should be replaced with ‘its effect on

 central nervous system (CNS)’

6

Abstract, line 21

PIP

Should be replaced with’its’

7

Abstract, line 21

com-petitive

competitive

8

Abstract, line 22

xenobi-otics

xenobiotics

9

Abstract, line 23

tech-nology

 technology

10

Abstract, line 25

dose

doses

11

Abstract, line 26

me-tabolites

metabolites

12

Abstract, line 28

synthe-size

synthesize

Abstract, line 28

and

Replace “with”

Abstract, line 28,29

this study aims to synthe-size current knowledge of PIP pharmacology and biochemistry and relate them to neurodegenerative and neurological disease therapy.

 Can be reworded as “this study aims to synthesize current knowledge of PIP pharmacology, biochemistry and relate them to alternative therapeutic interventions for neurodegenerative and neurological diseases.”

13

Introduction, line35

isolat-ed

isolated

14

Introduction, line39

re-ported

reported

15

Introduction, line40

meta-bolic

metabolic

16

Introduction, line50

pa-tients

patients

17

Introduction, line52

Never-theless

Nevertheless

18

Introduction, line54

im-proved

improved

19

Introduction, line57

sev-eral

several

20

Introduction, line58

phar- ma-cokinetic

phar- macokinetic

21

Introduction, line61

formu-lations

formulations

22

Pharmacokinetic and pharmacodynamic properties of PIP, line74

at

in

  1. All scientific names should be italics e.g., page line 35, Piper nigrum should be “Piper nigrum.”
  2. Table 1: Add caption below table.
  3. Table 1: What the authors wants to convey by “estimated amount (%)”. Is it dry weight or wet weight percentage?
  4. In subsection 4, a mechanistic figure can be included to make the content clearer.
  5. Following very important references should be added.
  6. Rehman MU, Rashid S, Arafah A, et al. Piperine Regulates Nrf-2/Keap-1 Signalling and Exhibits Anticancer Effect in Experimental Colon Carcinogenesis in Wistar Rats. Biology (Basel). 2020;9(9):302.
  7. Adil Farooq Wali, Shafat Ali, Summya Rashid, Rana M. Alsaffar, Azher Arafah, Wajhul Qamar, Ajaz Ahmad, Bilal Ahmad Paray, Sheikh Bilal Ahmad, Bashayr M. Alsuwayni, Muneeb U. Rehman, Attenuation of oxidative damage-associated hepatotoxicity by piperine in CCl4-induced liver fibrosis, Journal of King Saud University - Science, Volume 33, Issue 8, 2021, 101629.
  8. Vaibhav K, Shrivastava P, Javed H, et al. Piperine suppresses cerebral ischemia-reperfusion-induced inflammation through the repression of COX-2, NOS-2, and NF-κB in middle cerebral artery occlusion rat model. Mol Cell Biochem. 2012;367(1-2):73-84.
  9. Khan A, Jahan S, Imtiyaz Z, Alshahrani S, Antar Makeen H, Mohammed Alshehri B, Kumar A, Arafah A, Rehman MU. Neuroprotection: Targeting multiple pathways by naturally occurring phytochemicals. Biomedicines. 2020 Aug;8(8):284.

 On the basis of the above comments, I recommend the major changes before acceptance.

Author Response

Overall and general recommendation

The concept of the study is not novel but the authors have tried to justify with the title. My major concern about this review is that the authors have discussed only four diseases out of which, three are neurodegenerative diseases. The authors have discussed only one neurological disease i.e., Epilepsy. So, the title is not fully satisfied by the content. My suggestion is that either the title of the manuscript should be modified or the content should include more neurological diseases.

Response: We appreciate this concern. Our manuscript includes three most prevalent neurodegenerative diseases and few more neurological diseases including epilepsy, depression, fragile X-associated tremor/ataxia syndrome (FXTAS) and anxiety. Besides, we also have discussed about PIP role in neuroinflammation, which is most common feature for many neurodegenerative and neurological diseases.

Other major issues which needs attention are:

  1. The manuscript also has multiple grammatical mistakes which I have listed in the table below:

Serial no

Section/Location

Existing text

Comments/proposed text

Author response

1

Abstract, line 14

sug-gesting

suggesting

Corrected

2

Abstract, line 14

target

should be replaced with ‘intervention’

Replaced

3

Abstract, line 16

neuro-degenarative

neurodegenarative

Corrected

4

Abstract, line 17

compound important study material

 Should be replaced with ‘important investigational compound’

Corrected

5

Abstract, line 18,19

 effect in central nervous system (CNS)

Should be replaced with ‘its effect on

 central nervous system (CNS)’

Replaced

6

Abstract, line 21

PIP

Should be replaced with’its’

Replaced

7

Abstract, line 21

com-petitive

competitive

Corrected

8

Abstract, line 22

xenobi-otics

xenobiotics

Corrected

9

Abstract, line 23

tech-nology

 technology

Corrected

10

Abstract, line 25

dose

doses

Corrected

11

Abstract, line 26

me-tabolites

metabolites

Corrected

12

Abstract, line 28

synthe-size

synthesize

Corrected

Abstract, line 28

and

Replace “with”

Replaced

Abstract, line 28,29

this study aims to synthe-size current knowledge of PIP pharmacology and biochemistry and relate them to neurodegenerative and neurological disease therapy.

 Can be reworded as “this study aims to synthesize current knowledge of PIP pharmacology, biochemistry and relate them to alternative therapeutic interventions for neurodegenerative and neurological diseases.”

Corrected

13

Introduction, line35

isolat-ed

isolated

Corrected

14

Introduction, line39

re-ported

reported

Corrected

15

Introduction, line40

meta-bolic

metabolic

Corrected

16

Introduction, line50

pa-tients

patients

Corrected

17

Introduction, line52

Never-theless

Nevertheless

Corrected

18

Introduction, line54

im-proved

improved

Corrected

19

Introduction, line57

sev-eral

several

Corrected

20

Introduction, line58

phar- ma-cokinetic

phar- macokinetic

Corrected

21

Introduction, line61

formu-lations

formulations

Corrected

22

Pharmacokinetic and pharmacodynamic properties of PIP, line74

at

in

Replaced

  1. All scientific names should be italics e.g., page line 35, Piper nigrum should be “Piper nigrum.”

Response: Thank you for this suggestion, we have revised as per suggestion.

  1. Table 1: Add caption below table.

Response: We have added a caption below the table.

  1. Table 1: What the authors wants to convey by “estimated amount (%)”. Is it dry weight or wet weight percentage?

Response: The estimated amounts were calculated on dry weight basis, which we wrote in the caption.

  1. In subsection 4, a mechanistic figure can be included to make the content clearer.

Response: Thank you for this suggestion, as we mentioned in the final paragraph of future perspective section that PIP has not been fully studied in neurological or neurodegenerative models. It would be difficult to draw PIP molecular mechanisms in neurological or neurodegenerative diseases.

  1. Following very important references should be added.
  2. Rehman MU, Rashid S, Arafah A, et al. Piperine Regulates Nrf-2/Keap-1 Signalling and Exhibits Anticancer Effect in Experimental Colon Carcinogenesis in Wistar Rats. Biology (Basel). 2020;9(9):302.
  3. Adil Farooq Wali, Shafat Ali, Summya Rashid, Rana M. Alsaffar, Azher Arafah, Wajhul Qamar, Ajaz Ahmad, Bilal Ahmad Paray, Sheikh Bilal Ahmad, Bashayr M. Alsuwayni, Muneeb U. Rehman, Attenuation of oxidative damage-associated hepatotoxicity by piperine in CCl4-induced liver fibrosis, Journal of King Saud University - Science, Volume 33, Issue 8, 2021, 101629.
  4. Vaibhav K, Shrivastava P, Javed H, et al. Piperine suppresses cerebral ischemia-reperfusion-induced inflammation through the repression of COX-2, NOS-2, and NF-κB in middle cerebral artery occlusion rat model. Mol Cell Biochem. 2012;367(1-2):73-84.
  5. Khan A, Jahan S, Imtiyaz Z, Alshahrani S, Antar Makeen H, Mohammed Alshehri B, Kumar A, Arafah A, Rehman MU. Neuroprotection: Targeting multiple pathways by naturally occurring phytochemicals. Biomedicines. 2020 Aug;8(8):284.

Response: Thank you for suggesting very important papers. We have added c & d from the suggested list. As a & b discussed about PIP role in cancer models, we could not relate their findings with our manuscripts aim. Our manuscript mostly focusing on neurodegenerative or neurological models’ perspective. Thanks again for this generous suggestion.

Round 2

Reviewer 2 Report

R22- ”mono amino oxide B (MAO-B)”- Please replace with ”monoamine oxidase B”

R 136 PIP-mediated regulatin of drug pharmacology- Please replace ….regulation…

R 438 The major limitation of current study is absent of solution of PIP toxicology.  Please rephrase  this sentence

Author Response

R22- ”mono amino oxide B (MAO-B)”- Please replace with ”monoamine oxidase B”

Response: Thank you for the corrections, we have made the change in revised version.

R 136 PIP-mediated regulatin of drug pharmacology- Please replace ….regulation…

Response: Sorry for the typo mistake, we have corrected in revised version.

R 438 The major limitation of current study is absent of solution of PIP toxicology.  Please rephrase  this sentence

Response: Appreciate this suggestion, we have rephrase the sentence.